# Imageological/Structural Study regarding the Improved Pharmacokinetics by ^68^Ga-Labeled PEGylated PSMA Multimer in Prostate Cancer

**DOI:** 10.3390/ph16040589

**Published:** 2023-04-14

**Authors:** Huihui Zhang, Maohua Rao, Huayi Zhao, Jianli Ren, Lan Hao, Meng Zhong, Yue Chen, Xia Yang, Yue Feng, Gengbiao Yuan

**Affiliations:** 1Department of Nuclear Medicine, The Second Affiliated Hospital of Chongqing Medical University, Chongqing 400010, China; 2Chongqing Key Laboratory of Ultrasound Molecular Imaging, Second Affiliated Hospital of Chongqing Medical University, Chongqing 400010, China; 3Department of Pharmacy, The Affiliated Hospital of Southwest Medical University, Luzhou 646600, China; 4Nuclear Medicine and Molecular Imaging Key Laboratory of Sichuan Province, Luzhou 646600, China; 5Institute of Nuclear Physics and Chemistry, Academy of Engineering Physics, Mianyang 621900, China

**Keywords:** PSMA, prostate cancer, PSMA dimer, molecular probe, PET imaging

## Abstract

PMSA (prostate-specific membrane antigen) is currently the most significant target for diagnosing and treating PCa (prostate cancer). Herein, we reported a series ^68^Ga/^177^Lu-labeled multimer PSMA tracer conjugating with PEG chain, including [^68^Ga]Ga-DOTA-(1P-PEG_4_), [^68^Ga]Ga-DOTA-(2P-PEG_0_), [^68^Ga]Ga-DOTA-(2P-PEG_4_), and [^68^Ga]Ga/[^177^Lu]Lu-DOTA-(2P-PEG_4_)_2_, which showed an advantage of a multivalent effect and PEGylation to achieve higher tumor accumulation and faster kidney clearance. To figure out how structural optimizations based on a PSMA multimer and PEGylation influence the probe’s tumor-targeting ability, biodistribution, and metabolism, we examined PSMA molecular probes’ affinities to PC-3 PIP (PSMA-highly-expressed PC-3 cell line), and conducted pharmacokinetics analysis, biodistribution detection, small animal PET/CT, and SPECT/CT imaging. The results showed that PEG_4_ and PSMA dimer optimizations enhanced the probes’ tumor-targeting ability in PC-3 PIP tumor-bearing mice models. Compared with the PSMA monomer, the PEGylated PSMA dimer reduced the elimination half-life in the blood and increased uptake in the tumor, and the biodistribution results were consistent with PET/CT imaging results. [^68^Ga]Ga-DOTA-(2P-PEG_4_)_2_ exhibited higher tumor-to-organ ratios. When labeled by lutetium-177, relatively high accumulation of DOTA-(2P-PEG_4_)_2_ was still detected in PC-3 PIP tumor-bearing mice models after 48 h, indicating its prolonged tumor retention time. Given the superiority in imaging, simple synthetic processes, and structural stability, DOTA-(2P-PEG_4_)_2_ is expected to be a promising tumor-targeting diagnostic molecular probe in future clinical practice.

## 1. Introduction

According to the global statistical data regarding cancers issued by WHO in 2020, PCa (prostate cancer) has become the second largest threat to male health among malignant tumors [1], and ranks second only to lung cancer. In recent years, the incidence of PCa has been increasing rapidly. People with PCa are mostly diagnosed at the middle and advanced stages, severely impacting patients’ prognosis and life quality. The key to treating PCa falls on early diagnosis. PSMA is a type Ⅱ integrated membrane glycoprotein consisting of 750 amino acids [2]. Due to the significantly high expression in PCa, which is 100~1000 times higher than that in normal cells, PSMA has been regarded as an ideal biomarker for PCa diagnosis and therapy [3].

PSMA small-molecule inhibitors applied in the clinic are usually labeled by radionuclide fluorine-18 and gallium-68 as imaging agents, such as [^68^Ga]Ga-PSMA-11 [4,5], [^68^Ga]Ga-PSMA-617 [6], [^68^Ga]Ga-PSMA-I&T [7], [^18^F]F-PSMA-1007 [8], and [^18^F]F-DCFPyL [9], among which the FDA has approved [^68^Ga]Ga-PSMA-11 and [^18^F]F-DCFPyL in the past two years. There are two radiopharmaceuticals commonly applied in the clinic, namely [^177^Lu]Lu-PSMA-617 and [^177^Lu]Lu-PSMA-I&T [10]. However, they both inevitably exhibit high accumulation in non-targeting organs, including salivary glands, kidneys, intestines, etc. Meanwhile, approximately 30% of patients gain unsatisfactory therapeutic effects and their hematotoxicity limits their clinical application [11,12,13,14,15,16,17]. These problems could be due to the decreased effective dose in tumor cells caused by low ligand targeting ability and poor pharmacokinetics.

Researchers have proposed several optimization strategies regarding the pharmacokinetics of PSMA radioactive ligands to improve the tumor-targeting ability and settle the insufficiency of pharmacokinetics. Strategies included extending in vivo retention time by adding albumin-binding groups ((P-iodophenyl)butyric acid, ibuprofen, Evans Blue, etc. [15,18,19,20]) to prolong the serum half-life of drugs, or reorganizing DOTA chelating agent to enhance the binding between ligand and receptor [21], and reduced toxicity towards non-targeting organs and tissues [22]. Hence, a drug-designing scheme that endows PSMA with improved tumor-targeting ability, reduced non-targeting accumulation, accelerated clearance, and lessened toxicity to normal tissues is urgently required. 

According to previous research, polypeptides possess more effective ligand-binding force and superior pharmacokinetics due to their multivalent effect, which could improve tumor-targeting ability [23,24]. PEG has been ubiquitously applied to improve pharmacokinetics [16,25,26,27]. To prolong tumor retention time and improve pharmacokinetics, based on the multivalent effect of PSMA, a series of PSMA small molecules with different valence states were successfully designed and synthesized, followed by structural optimization by PEG_4_, namely DOTA-(1P-PEG_4_), DOTA-(2P-PEG_0_), DOTA-(2P-PEG_4_), and DOTA-(2P-PEG_4_)_2_ (Figure 1). These molecular probes’ corresponding tumor-targeting ability, pharmacokinetics, and tumor retention time were verified in vitro and in vivo by ^68^Ga/^177^Lu-labeling. In our preliminary research findings, the PSMA dimer showed a higher tumor accumulation rate in PCa than the PSMA monomer. The PEG_4_ chain not only improved the tumor uptake of ligands by prolonging serum half-life, but also accelerated drug metabolism and kidney clearance. Although we have not entered into the research of DOTA-(2P-PEG_4_)_2_-based PSMA TRT (targeting radionuclide therapy), [^177^Lu]Lu-DOTA-(2P-PEG_4_)_2_ still shows high tumor accumulation after 48 h, showing great potential for therapeutic applications in PCa.

## 2. Materials and Methods

### 2.1. General

All solvents and reagents were purchased from Sigma-Aldrich (Shanghai, China) and Shanghai yuanye Bio-Technology (Shanghai, China). DOTA-(1P-PEG_4_), DOTA-(2P-PEG_0_), DOTA(DOTA-(2P-PEG_4_), and (DOTA-(2P-PEG_4_)_2_ were synthesized by Bidepharm (Shanghai, China), and characterized by HPLC and mass spectrometry. Radio-HPLC analyses of the radiotracers were carried out using C18 column (5 µm, 4.6 × 250 mm, ZORBAX Eclipse Plus, Agilent, Milford, MA, USA), and analytical grade solvent by solvent A (0.1% trifluoroacetic acid (TFA) in water) and solvent B (0.1% TFA in acetonitrile). A ^68^Ge/^68^Ga generator was purchased from ITM Medical Isotopes (Munich, Germany). The γ count was measured by a CAPRAC-t γ counter containing a NaI crystal (CAPINTEC, Ramsey, NJ, USA). All in vitro experiments regarding PSMA and the determination of radiotracers’ stability were performed in triplicate to ensure reproducibility.

### 2.2. Chemistry and Radiochemistry

^68^GaCl^3+^ was eluted from a ^68^Ge/^68^Ga generator with 0.05M HCl (≈200 MBq/mL). A total of 20 nmol DOTA-(1P-PEG_4_), DOTA-(2P-PEG_0_), DOTA-(2P-PEG_4_), or DOTA-(2P-PEG_4_)_2_ was added into 1 mL of the eluted ^68^GaCl_3_ solution, followed by regulating pH to 4.5–5 with sodium acetate solution (0.5 M). The reaction system was sealed after shaking evenly and heated at 95 °C for 15 min, after which ^68^Ga-labeled DOTA-PSMA-PEG compounds were produced. The radiochemical purity and in vitro stability of the reaction products (20 µL) were examined by HPLC and thin-layer chromatography with the mobile phase of 0.1 M tri-sodium-citrate to a PH of 5.0. The radiolabeled ligands remained at the origin while the free gallium-68 moved with the solvent front by radio-TLC. The same labeling method and analysis method was applied to lutetium-177.

### 2.3. Cell Culture

Cellular experiments in this study were performed in the PC-3 PIP cell line. PC-3 PIP, a PCa cell line with stably high-expressed PSMA [28], was a gift from Professor Chen from the National University of Singapore. Cells were cultured in ATCC 1640 medium with 10% of FBS, L-glutamine, and antibiotic (37 °C, 5% CO_2_).

### 2.4. In Vitro Internalization Assay

PC-3 PIP cells were cultured in ATCC 1640 medium containing 10% FBS. When cultured to 70–80% confluency, cells were collected and seeded in a 24-well plate. After culturing overnight (37 °C, 5% CO_2_), the culture medium was replaced by serum-free ATCC 1640 medium (serum-starved cell culture) for 0.5 h. Then nuclide-labeled compounds were added into wells (50 µL, 1 μCi/well) with gentle shakes to ensure they were well distributed. After incubating for 30 min/60 min/120 min, cells at each time point were digested by NaOH and measured by the γ-counter to calculate the uptake rate (*n* ≥ 3). Cells were co-incubated with glycine (pH = 2.8) for 1 min, and then the procedure above was repeated after dissolving in 1 M NaOH (procedure for internalization). Cells were incubated with PSMA inhibitor (2-PMPA) for blocking assays for 15 min before adding the labeled ligands. 

### 2.5. Lipophilicity Determination

Nuclide-labeled ligand (20 μL, ≈50 μCi), PBS (480 μL), and saturated n-octanol (500 μL) were added in a 1 mL EP tube, followed by vortex. Then the mixture was centrifuged at a low temperature (−5 °C) for 10 min until the mixture was separated into the upper and the lower layers. Liquids in the two phases were transferred into EP tubes and measured by the γ-counter. The calculation of the LogP value was based on the following equation:

LogP = Lg [(counting of the organic phase − background counting)/(counting of the aqueous phase − background counting)]

### 2.6. In Vitro Determination of Albumin Binding

The albumin-binding rate of each ligand was measured by the γ-counter [16]. ^68^Ga-labeled PSMA ligand (≈50 μCi) was added into the human albumin solution (20 mg/mL). The mixture was incubated in a water bath (37 °C) for 30 min and 120 min, followed by ultrafiltration centrifugation for 5 min (10 kD, 10,000 rpm). The compound activities of the uncombined albumin or those retained on the filter membrane were measured by the γ counter, according to which the percentage of serum albumin binding activity (retained on the filter membrane) to total activity was calculated.

The calculation was based on the following equation: (counting of albumin retained on membrane − background counting)/(counting of albumin retained on membrane + counting of filtered liquid − 2* background counting) × 100%, and the average of three sets of data was taken as the final result.

### 2.7. Determination of the IC_50_ Values in PC3 PIP Cells 

PC-3 PIP cells were seeded in 96-well plates in triplicate and cultured to a density of 5* × 10^5^ cells per well. [^68^Ga]Ga-PSMA-11 was added into wells after serum-starved cell culture (1 μCi/well, ≈171 nmol/L), after which different PBS-diluted competitive ligands were added in wells and cultured for another hour. Then cells were washed with cold PBS twice removing the supernatant. Cells were collected and transferred to test tubes for counting by the γ-counter. IC_50_ values were analyzed and confirmed by nonlinear regression on GraphPad Prism Software.

### 2.8. Animal Tumor Model

PC-3 PIP cells at the logarithmic phase were collected after trypsinization and re-suspended to 5 × 10^7^/mL with serum-free medium after removing the supernatant. Mice (6 weeks) were subcutaneously injected with 200 μL of cell suspension, and utilized for small animal PET/CT imaging, SPECT/CT imaging, and biodistribution analysis when tumor volume was ≧100 mm^3^. All animal experiments were conducted with the approval of The Southwest Medical University Animal Ethics Committee.

### 2.9. Pharmacokinetics and Biodistribution in Mice

Healthy male SD rats were intravenously injected with four ^68^Ga-labeled ligands through the caudal vein. Blood was collected and weighed through the ocular venous plexus at different times. The radioactivity of the collected blood was measured by the γ counter, and the calculated results were presented as % ID/g. The blood pharmacokinetics of four ligands were evaluated using biventricular models. Blood metabolism models were established on the basis of Ct = A e^−αt^ + Be^−βt^ [29] after fitting ID%/g and time. 

PC-3 PIP tumor-bearing mice (*n* > 3) were divided into four groups and respectively injected with 0.1 mL (70μCi) of [^68^Ga]Ga-DOTA-(1P-PEG_4_), [^68^Ga]Ga-DOTA-(2P-PEG_0_), [^68^Ga]Ga-DOTA-(2P-PEG_4_), and [^68^Ga]Ga-DOTA-(2P-PEG_4_)_2_ through the caudal vein. Mice were executed at 30 min/60 min/120 min after injection. Heart, blood, lung, liver, stomach, spleen, kidney, muscle, intestine, bone, salivary gland, and tumor were collected and detected by the γ-counter. The fitted curve was calculated on the basis of standard diluted samples of the initial dosage, and then the percentage of injected dose per gram of tissue (ID%/g) was calculated.

### 2.10. In Vivo PET/CT and SPECT/CT Imaging

Tumor-bearing mice (*n* = 3) were intravenously injected with 0.1 mL (80μCi) of freshly prepared ^68^Ga/^177^Lu-labeled PSMA ligands. PET/CT imaging was performed at 0.5/1/2/5 h after injection, and SPECT/CT imaging was performed at 6/24/48 h after injection. The acquired PET/CT and SPECT/CT data were re-constructed by OSEM (ordered subsets expectation maximization).

### 2.11. Statistical Analysis

Relevant measurement data were proceeded by Prism 8.0.1. The student’s *t*-test was adopted to analyze measurement data. The difference was considered statistically significant when *p* < 0.05.

## 3. Results

### 3.1. Radiochemical Synthesis and Quality Control

A total of four kinds of PSMA-related ligands were designed and synthesized. The radiochemical purity of these four ^68^Ga/^177^Lu-labeled compounds was detected by HPLC (Figure 1) and thin-layer chromatography (Appendix A). The purity of each ^68^Ga-labeled compound was over 95% at a molar activity of up to 10 MBq/nmol (Figure 1A); the radiochemical purity of [^177^Lu]Lu-DOTA-(2P-PEG_4_)_2_ reached 99.89% (Figure 1B). The stabilities of [^68^Ga]Ga-DOTA-(1P-PEG_4_), [^68^Ga]Ga-DOTA-(2P-PEG_4_), [^68^Ga]Ga-DOTA-(2P-PEG_0_), and [^68^Ga]Ga-DOTA-(2P-PEG_4_)_2_ were examined after incubating in PBS/FBS at 37 °C for 2 h. As shown in Figure 1C,D, the radiochemical purity (PBS/FBS) of the four ligands at 2 h were 96.97/95.15%, 98.06/95.95%, 95.78/95.21%, and 99.55/98.99%, respectively. These results demonstrated that these newly synthesized PSMA-targeted ligands possessed sound stabilities in vitro and were eligible for subsequent research.

### 3.2. Partition Coefficient 

The LogP values of [^68^Ga]Ga-DOTA-(1P-PEG_4_), [^68^Ga]Ga-DOTA-(2P-PEG_0_), [^68^Ga]Ga-DOTA-(2P-PEG_4_), and [^68^Ga]Ga-DOTA-(2P-PEG_4_)_2_ were −3.02, −2.84, −3.09, and −3.25, respectively (Figure 2). Data in Figure 2 illustrates that the PSMA monomer was hydrophilic, while the hydrophilic performance of the PSMA tetramer was significantly better with the augmentation of hydrophilia by polypeptide PEGylation.

### 3.3. In Vitro Testing of Binding to Albumin

The 0.5/2 h albumin-binding rates of four ligands were detected after co-incubation with human serum albumin and ultrafiltration centrifugation. The binding rates (0.5/2 h) of ^68^Ga-labeled DOTA-(1P-PEG_4_), DOTA-(2P-PEG_0_), DOTA-(2P-PEG_4_), and DOTA-(2P-PEG_4_)_2_ were 35.51%/32.85%, 54.84%/54.29%, 68.38%/66.97%, and 95.17%/94.72%, respectively, indicating that the introduction of dimerized PSMA affected the binding of albumin. Multiple research results have shown that covalent conjugation of activated PEG could extend the serum half-life of proteins or polypeptides, and enhance chemical and biological stability [26,30], which was also verified by the different binding rates of DOTA-(2P-PEG_0_) and DOTA-(2P-PEG_4_) in our research.

### 3.4. In Vitro Experiments

PC-3 PIP cells were incubated with ^68^Ga-labeled DOTA-(1P-PEG_4_), DOTA-(2P-PEG_0_), DOTA-(2P-PEG_4_), and DOTA-(2P-PEG_4_)_2_, and the γ-counter was applied to detect the cellular uptake rate (Figure 3A). The 2 h cellular uptake rate of DOTA-(2P-PEG_4_) (13.26 ± 0.02%) was obviously higher than that of DOTA-(1P-PEG_4_) (7.65 ± 3.12%) and DOTA-(2P-PEG_0_) (8.39 ± 3.65%), which was possibly due to enhanced stability by PEGylation and promoted affinity ability by the PSMA dimer. The 2 h uptake rate of the PMSA tetramer DOTA-(2P-PEG_4_)_2_ reached 27.57 ± 7.07%, and both DOTA-(2P-PEG_4_)_2_ and DOTA-(2P-PEG_4_) had higher cellular uptake rates than DOTA-(1P-PEG_4_), proving that the multimer PSMA could benefit the binding between ligands and tumor cells. After PC-3 PIP cells were co-incubated with PSMA inhibitor 2-PMPA, the uptake rates of [^68^Ga]Ga-DOTA-(1P-PEG_4_), [^68^Ga]Ga-DOTA-(2P-PEG_0_), [^68^Ga]Ga-DOTA-(2P-PEG_4_), and [^68^Ga]Ga-DOTA-(2P-PEG_4_)_2_ significantly decreased to 0.51 ± 0.14%, 1.00 ± 0.16%, 1.35 ± 0.36%, and 0.91 ± 0.19% (*p* < 0.0001), respectively, indicating the PSMA ligands mentioned above possessed good targeting ability towards PSMA-positive PC-3 PIP cells.

According to the IC_50_ values in Figure 3D, all ligands possessed good affinities within the nanomole range, where the IC_50_ values of [^68^Ga]Ga-DOTA-(1P-PEG_4_),[^68^Ga]Ga-DOTA-(2P-PEG_0_), [^68^Ga]Ga-DOTA-(2P-PEG_4_), and [^68^Ga]Ga-DOTA-(2P-PEG_4_)_2_ were 793.5 nM, 454.6 nM, 242.3 nM, and 42.40 nM, respectively, indicating that both PEGylation and polymerized PSMA improved ligand receptor binding affinity, while the multivalent effect made it even better.

### 3.5. Pharmacokinetics in Blood and Biodistribution

SD rats were randomly divided into four groups and intravenously injected with ^68^Ga-labelled DOTA-(1P-PEG_4_), DOTA-(2P-PEG_0_), DOTA-(2P-PEG_4_), and DOTA-(2P-PEG_4_)_2_. The relation curve between average plasma concentration and time was plotted. As shown in Figure 4, the concentrations of all four compounds declined in a multi-exponential manner after injection. According to detection and calculation, the AUC_0→4_ h values indicated approximately a two-fold increased blood retention for [^68^Ga]Ga-DOTA-(2P-PEG_4_)_2_ compared with [^68^Ga]Ga-DOTA-(1P-PEG_4_). The pharmacokinetics of DOTA-(1P-PEG_4_), DOTA-(2P-PEG_0_), DOTA-(2P-PEG_4_), and DOTA-(2P-PEG_4_)_2_ in the blood accorded with following equations, respectively: Ct _[68Ga]Ga-DOTA-(1P-PEG4)_ = 0.787e^−11.434t^ + 0.382e^−1.949t^,(1)
Ct _[68Ga]Ga-DOTA-(2P-PEG4)_ = 1.529e^−33.186t^ + 0.538e^−1.220t^,(2)
Ct _[68Ga]Ga-DOTA-(2P-PEG0)_ = 0.882e^−9.017t^ + 0.197e^−0.834t^,(3)
Ct _[68Ga]Ga-DOTA-(2P-PEG4)2_ = 1.832e^−13.798t^ + 0.438e^−1.194t^
(4)

To elucidate the impact of PEG-modified polymerized PSMA ligands on tumor uptake and metabolism, mice were executed after injecting ^68^Ga-labeled ligands to detect the biodistribution in major organs and tumors. As shown in Figure 5A–C, all four ligands could obviously accumulate in PSMA-positive tumors, among which DOTA-(2P-PEG_4_)_2_ exhibited the highest tumor uptake value, maintaining over 20ID%/g, and this value rose to 21.85 ± 3.53ID%/g at 2 h. DOTA-(2P-PEG_4_)_2_ kidney uptake at 30 min reached the peak and was obviously higher than that of other ligands, followed by rapidly decreasing to 4.76 ± 2.63ID%/g at 2 h, which was significantly lower than that of DOTA-(1P-PEG_4_) (Figure 5C, Appendix A). We speculated that the symmetric PSMA dimer-PEG_4_ accounted for this result. The symmetric PEG_4_ enhanced the hydrophilia of DOTA-(2P-PEG_4_)_2_, which accelerated kidney clearance. The crafty designing of the PSMA tetramer promoted tumor-targeting ability, and avoided impeding the binding site by long-chain PEG [31]. Compared with the commonly used drug reported by other researchers, such as PSMA-11 [32] and PSMA-617 [13], the optimized four ligands exhibited less uptake in the salivary gland and gastrointestinal tract, among which DOTA-(2P-PEG_4_)_2_ was detected to have the lowest salivary gland uptake at 2 h (0.17 ± 0.03ID%/g). The 2 h blood uptake values of DOTA-(1P-PEG_4_) and DOTA-(2P-PEG_0_) (6.11 ± 4.84 ID%/g, 2.78 ± 1.80 ID%/g) were remarkably higher than that of DOTA-(2P-PEG_4_) and DOTA-(2P-PEG_4_)_2_ (0.69 ± 0.28 ID%/g, 1.14 ± 1.43 ID%/g), which may be due to the relatively lower albumin-binding rate of DOTA-(1P-PEG_4_). DOTA-(2P-PEG_0_) was not qualified for therapeutic research, considering that it was unstable without PEGylation, possibly fortifying blood distribution and increasing hematotoxicity.

### 3.6. Imaging of Micro Animal PET/CT and SPECT/CT

The Micro Animal PET/CT and SPECT/CT imaging of mice injected with ligands are presented in Figure 6. The results indicated that all the radiotracers were absorbed by the tumor, and other organs demonstrated no obvious specific uptake except for the kidney and the bladder. The tumor uptake of DOTA-(1P-PEG_4_), DOTA-(2P-PEG_0_), and DOTA-(2P-PEG_4_) began to decline at 2 h after injection (Figure 6A–C), while DOTA-(2P-PEG_4_)_2_ remained stable and exhibited obvious uptake in the tumor at 5 h after injection (Figure 6D). Furthermore, DOTA-(2P-PEG_4_)_2_ exhibited a favorable T/NT ratio at all time points, with a remarkably lower kidney uptake than [^68^Ga]Ga-PSMA-11 (Figure 6E). DOTA-(2P-PEG_4_)_2_ was further labeled by lutetium-177 and utilized for imaging. [^177^Lu]Lu-DOTA-(2P-PEG_4_)_2_ showed better imaging capacity than [^177^Lu]Lu-PSMA-617 (Figure 6G), and the in vivo tumor uptake was still obvious at 48 h after injection (Figure 6F), indicating that it possessed a longer detention time in vivo and a sound tumor-targeting ability. Conclusively, DOTA-(2P-PEG_4_)_2_ was a potential candidate drug for radionuclide therapy. In addition, the tumor uptake was apparently blocked in a 2-PMPA-blocking experiment (Figure 6H), indicating that these ligands had great PSMA specificity (Figure 6H).

## 4. Discussion

Polymeric peptides constructed on the basis of the polyvalent effect can improve the tumor-targeting ability and prolong tumor retention time remarkably. Plenty of polymerized peptides have been reported previously, and polymerization can be performed between the same peptides, such as 2P(RGD)_2_ [33] and 2P(FAPI) [24], or between two kinds of peptides, for example, PSMA-FAPI [34] and RGD-FAPI [35]. According to current clinical data, based on polymerization strategy, 2P(RGD)_2_ and 2P(FAPI) have been constructed and testified to gain an increased tumor uptake rate and better tumor retention capacity. In this study, a polymeric modification was performed on PSMA, by which we obtained a series of polymerized molecules, namely DOTA-(1P-PEG_4_), DOTA-(2P-PEG_0_), DOTA-(2P-PEG_4_), and DOTA-(2P-PEG_4_)_2_, and conducted particular in vitro and in vivo assessments and examinations on these ^68^Ga-labeled PSMA ligands. According to the results of the in vitro experiments, these newly synthesized radiotracers possessed sound stability, and the monomer, dimer, and tetramer all obtained superior binding capacity, among which the tetramer achieved the best effect. Phenomena observed by PET imaging and biodistribution analysis proved that these ligands could be metabolized rapidly in vivo, while signals of ligands in tumors were still detectable at 5 h after injection, where the uptake of the tetramer remained the highest.

The introduction of PEG chains is generally recognized as a way to optimize pharmacokinetics [36], which has been applied in designing many radiotracers. Previous studies have reported that replacing PEG_3_ with PEG_3_ + PEG_7_ could improve tumor uptake, prolong tumor retention time, and reduce kidney uptake [16]. Nonetheless, some reports concluded that tumor uptake and biodistribution were not improved when PEG_4_ had a modified structure, especially when the PEG chain was too long; for example, the modification of PEG_12_/PEG_24_ markedly impeded the affinity of the compound [27,37], which could be caused by the “PEG dilemma” [38]. Other research reports that PEG_2_/PEG_4_ modification can maintain the maximum affinity of PSMA [31]. We explored the effects of PEG_4_ on the affinity of PSMA ligands. DOTA-(2P-PEG_4_) was constructed by insert a PEG_4_ chain into DOTA-2P, and it exhibited better tumor uptake according to imaging results, indicating that PEG_4_ slightly facilitated the compound’s affinity. In addition, DOTA-(2P-PEG_4_) possessed better stability compared with DOTA-(2P-PEG_0_). From the aspect of molecular structure, PEG_4_ endowed the molecule with extra flexibility [38], which enabled the binding between ligand and binding spot, as well as maintaining biostability. Although the DOTA-(2P-PEG4) reduced kidney uptake to some extent, referring to our imaging data and cellular experiment results, improvement in the tumor-targeting ability of DO-TA-(2P-PEG4) was not acquired. Nevertheless, the effect of DOTA-(2P-PEG_4_) was not as satisfactory as we expected; although kidney uptake was reduced to some extent, referring to our imaging data and cellular experiment results, improvement in the tumor-targeting ability of DOTA-(2P-PEG_4_) was not acquired.

To address the low tumor uptake of the PSMA dimer, we designed DOTA-(2P-PEG_4_)_2_, which again adopted a polymeric peptide strategy, aiming to enhance the tumor-targeting ability of the symmetric PSMA dimer. Meanwhile, the PEG_4_ was doubled, which further accelerated kidney clearance. Some reports claim that the peptide tetramer and peptide octamer can increase tumor uptake, but, more significantly, the kidney uptake [39]. To the best of our knowledge, we have found that PEG chains could improve the pharmacokinetics, and have the potential to overcome this problem. Fortunately, imaging and cellular experiments successfully validated this deduction of our newly designed structures. The cellular affinity of DOTA-(2P-PEG_4_)_2_ was 4–5 times more than that of DOTA-(2P-PEG_4_), and the uptake rate of DOTA-(2P-PEG_4_)_2_ was 2–3 times higher than that of DOTA-(2P-PEG_4_). In the imaging experiment, the tumor uptake of DOTA-(2P-PEG_4_)_2_ was higher than that of DOTA-(2P-PEG_4_) at each time point, and the kidney uptake of DOTA-(2P-PEG_4_)_2_ at 2 h was basically undetectable. We realized the 1 + 1 > 2 effect through this crafty design.

Serum retention time is also an essential factor in radiopharmaceuticals. In this study, we found that the albumin-binding rate could be promoted by simply increasing PSMA numbers, since the albumin-binding rate of DOTA-(2P-PEG_4_)_2_ reached 94.7%, which has not been reported by any other researchers before. This phenomenon was possibly caused by the fatty acid of glutamic acid in the PSMA molecule binding to the site2 in albumin [40]. The albumin-binding rate was improved with increasing glutamic acid numbers in the symmetric PSMA dimer. In the pharmacokinetics experiment, the serum retention time of the tetramer was short, which may have been due to the PEG_4_ chain, considering that PEG_4_ could significantly accelerate serum clearance and kidney clearance.

To explore the therapeutic potential, DOTA-(2P-PEG_4_)_2_ was labeled by lutetiun-177. According to imaging results, tumor uptake remained relatively high at 48 h, indicating that his polymerized PSMA possessed not only improved tumor-targeting ability, but also prolonged tumor retention time. Compared with DOTA-(2P-PEG_4_)_2_, [^177^Lu]Lu-PSMA-617 exhibited detectable kidney uptake at 6 h, while, at the same time, DOTA-(2P-PEG_4_)_2_ had no obvious signal detected except for tumor, showing the advantage of DOTA-(2P-PEG_4_)_2_ in reducing radiation damage to the kidney. This result was also verified by Micro Animal PET/CT imaging of ^68^Ga-labeled DOTA-(2P-PEG_4_)_2_, and PSMA-11; kidney clearance of [^68^Ga]Ga-DOTA-(2P-PEG_4_)_2_ was obviously accelerated. To sum up, DOTA-(2P-PEG_4_)_2_ has the potential to be applied as a promising therapeutic agent.

## 5. Conclusions

In this study, we firstly synthesized polymerized PSMA ligands, namely DOTA-(2P-PEG_0_), DOTA-(2P-PEG_4_), and DOTA-(2P-PEG_4_)_2_. The designed symmetric PSMA dimer significantly improved the affinity to the PSMA ligand and prolonged the detention time in the tumor. With the modification of the PEG chain, not only were the compounds’ stability, hydrophilia, and albumin-binding rate improved, but also the kidney uptake of ligands was significantly reduced. In conclusion, the modification strategies in this study may provide a fresh idea for optimizing PC-targeting PSMA molecular probes, and DOTA-(2P-PEG_4_)_2_ has shown potential in PSMA high expression tumor imaging and therapy.

## Data Availability

Data is contained within the article and Appendix A.

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
