# Peer review of "Imageological/Structural Study regarding the Improved Pharmacokinetics by 68Ga-Labeled PEGylated PSMA Multimer in Prostate Cancer"

_pharmaceuticals, 2023, doi:10.3390/ph16040589_

Round 1

Reviewer 1 Report

The manuscript 2290274 by Zhang et al present the preclinical evaluation and comparison of 4 PSMA analogues intended for diagnostic and therapeutic use. The study design is relevant and the experimental work well performed. The conclusions drawn are in line with the results and the overal interest for the reader is rather high. 

However, the language needs to be improved and a number of corrections and changes are required before this manuscript can be accepted. The number of detailed changes are too many to be pointed out but some suggestions are listed below.

Neither the radiochemical yield or the molar activity is reported. In the experimental section 4.2 Chemistry and radiochemistry, the amount of precursor used for the chelating reaction with gallium-68 or lutetium-177 is not mentioned. Figure 1, C and D report radiochemical yield (RCY) which should be radiochemical purity. The figure legend to Figure 1 is not correct "Stability and labeling rates...." and the mV values in A and B could be written i a different manner. I question that 4 significant figures in C and D is relevant. The reported value for lutetium-177 compound in B, 99.53%, cannot be correct considering the signal to noise in the chromatogram.  The radiochemical purity should also be shown with one more chromatographic method, TLC could be added.   

It is not alway clear if the experiment performed is with the ligand with our without gallium-68.  Two examples are, 2.3 In vitro testing and results in figure 4. Another is 4.7 Determination of IC50 values. The number of significant figures in this section such as 35.51% can be questioned. In section 2.4 line 146-147.....high affinity within low nanomole range.... but the values are definitely not low (793-40 nM). Table 1 should be in supplementary section. 

EDITORIAL

The text could be shorter and the language needs to be checked (line 60-69 is one example). The authors should follow the nomenclature guidelines for radiolabelled compounds (for example [68Ga]Ga-DOTA-[1P-PEG4), in text gallium-68. It would be simpler for the reader if this nomenclature was used throughout the manuscript to avoid misunderstanding if precursor peptide or the chelated labelled product is discussed.

Reviewer 2 Report

It is an interesting research in the preclinical field. The bibliography is varied and accurate.

However, I believe that some changes should be made and in particular: -line 43: “the decline of disease average age” should be better explained or, if not possible, eliminated

- the “Materials and methods” section in its entirety must precede section 2 (Results) as per the typical structure of a scientific article

Round 2

Reviewer 1 Report

A few spelling errors needs to be corrected. In text 68Ga should be written gallium-68.
